# Characterization of the Thermal Behavior of a Complex Composite (Clutch Facing) Combining Digital Image Stereo Correlation and Numerical Approach

**DOI:** 10.3390/ma15072582

**Published:** 2022-03-31

**Authors:** Camille Flament, Bruno Berthel, Michelle Salvia, Gérard Grosland, Isabelle Alix

**Affiliations:** 1Laboratoire de Tribologie et Dynamique des Systèmes, UMR CNRS 5513, Ecole Centrale de Lyon, Université de Lyon, 36 Avenue Guy de Collongue, 69134 Ecully, France; flament.clle@gmail.com; 2Valeo Matériaux de Friction, rue Barthélémy Thimonnier, 87020 Limoges, France; gerard.crosland@valeo.com (G.G.); isabelle.alix@valeo.com (I.A.)

**Keywords:** thermal expansion, DIC, fibre composite, FEMU, laminate theory, tomography

## Abstract

Car clutch facings are complex fiber-reinforced composites. The coefficient of thermal expansion (CTE) of the composite is one of the main thermal properties, which affects dry clutch engagement process due to heat associated with friction. In the case of clutch facing, which only exists in its final form as a non-planar annular disc, it is difficult to define an elementary representative volume. The objective of this work was to develop a method for identifying the CTE distributions on the entire part. A device allowing measuring the strain fields by digital image correlation (DIC) under homogeneous thermal loading (up to 300 °C) was developed. The experimental results highlight the heterogeneity and the orthotropic nature of the material behavior and the influence of the angle between the fibers on the CTE. To take into account that the measured strain fields are related to the CTE, but also to the shape of the part, different approaches to identify the CTE were considered: direct measurements, classical laminate theory (CLT) and finite element method updating (FEMU). Only the FEMU allows an accurate identification of the CTE distributions. Nevertheless, the CLT respects the orders of magnitude and remains a useful tool for the design of clutches.

## 1. Introduction

A car clutch is a device located between the engine and the gearbox. An organic clutch facing sandwiched between the flywheel and a pressure plate (cast iron) transmits the rotary motion between the engine and the wheels via the transmission. The current car clutch facings are made with phenolic-based binder reinforced principally with continuous glass, polyacrylonitrile (PAN) fibers, and copper strips. During dry clutch engagement, sliding contact occurs between the clutch facing and the counter faces. When sliding takes place, heat is generated due to dissipation mechanisms of kinetic energy. In the case of repetitive engagement, the temperature rise can be significant: up to 250 or 300 °C depending on clutch technology, which triggers thermal expansion of the material.

The design of a multi-material system subjected to thermal loading, such as a car clutch with imposed volume constraints, requires a suitable knowledge of the thermal expansion behavior of each material. The thermal expansion behavior is quantified through the fractional change in length under heating, or cooling, over a certain temperature range, i.e., the linear coefficient of thermal expansion (CTE). CTE is well known for the different parts of the clutch made of isotropic metallic material. For the composite facing, however, the CTE depends on several material parameters. Thermo-mechanical properties of fiber-reinforced composites (as clutch facing) are, in general, both anisotropic and non-homogeneous depending on matrix composition, reinforcement properties, and architecture (orientation, waviness, stacking sequence, etc.) and on the manufacturing process (non-planar part).

Several experimental techniques exist to measure thermal expansion coefficients in various environmental conditions (classical push-rod dilatometer [1,2,3,4], interferometry-based method [2], strain gauge [5,6], and optical fibers (OF) [7,8]). However, these techniques need specific sample preparation that can lead to errors (sample cutting, gauge or OF bonding or embedding, etc.) and only provide information on a specific local area where the sample was taken.

When working with inhomogeneous and/or anisotropic materials, such as composites, accurate strain measurements call for full-field data. Among the different full-field measurement techniques, the digital image correlation (DIC) method is one of the most popular. It is an optic contactless experimental technique to measure full-field displacements and strains with sub-pixel accuracy. This technique is used to obtain the mechanical and thermal properties of a wide range of materials [9,10,11,12,13,14,15,16,17,18,19]. In particular, the advantage of using this technique is that it is possible to analyze the local mechanical or thermal behavior of materials. When using one camera (2D-DIC), the temporal correlation takes only into account in-plane displacements. If the material surface is non-planar, or if out-of-plane displacement occurs during thermal or mechanical loading (as many composite materials), it is necessary to implement a stereo-vision system with two simultaneous images of the same object (DISC or 3D-DIC) with a different camera angle. The measured displacement fields then provide input data for the identification of the mechanical behavior of materials. In the case of complex geometry, the solution is to work with the complete structure, as in the work of [20]. The goal is to identify the material properties by imposing homogeneous boundaries conditions. The method consists of comparing displacement fields obtained by digital image correlation with those estimated numerically. This method, called finite elements model updating (FEMU), has been applied in a wide range of materials and loading [21,22,23].

Regarding prediction tools, classical laminate theory (CLT) is a simple and fast way to predict the behavior of composite laminates under different loading conditions (mechanical, thermal, or hygrothermal). The prediction requires knowledge of the properties of individual unidirectional layers obtained from the properties of its constituents (fiber and matrix) and their relative volume using micro-mechanical models (mixing rule) and the stacking sequence of the layers. In particular, it shows that the properties of an angle-ply laminate depend strongly on the angle of intersection of the plies [24]. When the properties of the plies as, for example, thermal expansion coefficients, are not available (e.g., due to the lack of knowledge of the properties of the constituents), their experimental measurements can be used as input data for the classical laminate theory calculations [25], which can then be used as a basis for composite design.

The aim of this study is to bring a better understanding of the thermal behavior of the composite constituting the clutch facings in order to provide material data necessary for accurate modeling of the assemblies to which it belongs. This work is, therefore, focused on the deformation behavior of the clutch facing material under the effect of temperature using experimental and numerical procedures.

In order to take into account the non-planarity and the complex organization of the reinforcements of the composite facing, an experimental setup based on 3D-DIC analysis was designed and validated. Thermal tests were performed on the overall “as received” facing versus use temperature range. The results were used to obtain the experimental coefficient of thermal expansion distribution of the material. Moreover, a simple numerical approach based on classical laminate theory (CLT) and the knowledge of the fiber organization on the one hand, and full-field measurements combined with the finite element method (FEM) on the other hand, were proposed to predict the coefficient of thermal expansion distribution in the case of a complex composite structure as clutch facing.

## 2. Materials and Methods

### 2.1. Clutch Facing Material Description

#### 2.1.1. Clutch Facing Material and Manufacturing Process

The organic clutch facing is an annular-shaped continuous fiber composite. The composite matrix is a mixture of phenolic resin, melamine formaldehyde resin, and vulcanized styrene-butadiene rubber (SBR). Three types of fibers are used: glass fibers, polyacrylonitrile (PAN) fibers, and copper fibers. The fibers are twisted into rovings and then impregnated with the resin mixture in a final proportion of 60% by volume. The diameter of the rovings is a few millimeters.

The main steps of the process are described in [26]. The yarn is impregnated with the resin and dried before being shaped into a preform (details are given in Section 2.1.2 and Section 2.1.3). Then, the preform is put into a heated mold, pressed, and then cured at 250 °C. The finished product has grooves and rivet holes. To better understand the structure, this work was conducted on specimens that were neither grooved nor drilled. The studied structure has external and internal diameters of 240 and 160 mm, respectively. It is 2.5 mm thick.

Due to the organization of the fibers, it can be assumed that the composite material used for the clutch facing has an orthotropic behavior. However, the mechanical and thermal properties of the individual components are not exactly known, as it is not possible to manufacture them separately. Thus, only an experimental approach can determine the mechanical and thermal properties of this composite material.

#### 2.1.2. Theoretical Fiber Organization

During the preforming operation, a machine guides the impregnated fibers coupling a uniform rotation with a radial translation. The two movements have different frequencies resulting in a fiber organization described in the Cartesian reference by Equation (1) and presented in Figure 1a:(1)$$\{\begin{array}{c} X(\theta )=\left[R_{0}+\frac{A_{0}}{2}\times sin\left(\theta \times L_{b}\right)\right]\times cos(\theta )\\ Y(\theta )=\left[R_{0}+\frac{A_{0}}{2}\times sin\left(\theta \times L_{b}\right)\right]\times sin(\theta ) \end{array}$$
where *R*_0_ is the mean radius, $A_{0}=\left(D_{out}-D_{in}\right)/2$ with *D_out_* and *D_in_* the outer and inner diameters, *L_b_* the preforming ratio, which is the number of sin waves per 2π phase angle. (*L_b_*) defines the fiber orientation.

Though the yarn is preformed in a continuous process, layers appear for each 2π phase. After locating the intersections, it is possible to estimate the angle at which the two layers cross by estimating the angle between the tangent of the curve and the radial direction (+α or −α) (Figure 1a). The angles at the intersections versus the radius are shown in Figure 1b. The calculation was done for *L_b_* and *A*_0_ = 40 mm (reference case).

This organization is close to that of an angle-ply laminate with the axes of the cylindrical coordinate system linked to the ring as orthotropic axes, with a gradient of change in the angle of intersection of the fibers. The equivalent laminate composite defined with rotational symmetry is given in Table 1. The average angle is 56°.

The actual fiber organization of the clutch facing is, in general, more heterogeneous as some movements may occur during preforming and curing, which can modify the impregnated yarn tracing. However, it can be used as a first approach to understanding the thermo-mechanical behavior of the finished structure.

#### 2.1.3. Experimental Fiber Organization

In order to validate the theoretical approach of the fiber organization as well as to better understand the relative position of each of the components of the composite, X-ray tomography images of the clutch facing were achieved. The tests were performed at the ESRF (European Synchrotron Radiation Facility, beamline: BM05, Grenoble, France) synchrotron. The detector is a CCD camera (FReLoN camera, “Analog and Transient Electronic” ESRF group, Grenoble, France) of 2048 × 2048 pixels, sensitive to X-rays. The energy of the incident beam was fixed at 120 keV. The integration time used was 0.12 s. A complete acquisition is made by a 360 °C rotation of the sample with 8000 2D projections taken. The conditions chosen allow a spatial resolution of 12 × 12 × 12 μm^3^.

Computed tomography is a non-destructive method based on the 3D X-ray scan, which allows three-dimensional structure information of materials without contact. It produces a series of greyscale images, representing sections of the volume in a given direction, which cover the entire volume. The gray levels are related to the density of the material through which the X-rays pass. It is thus possible to observe the relative position of the different constituents of a composite material if the spatial resolution and the density allow it.

The gray levels of the images produced by X-ray tomography are related to the density of the material. Dense materials tend to be white, while less dense materials, such as porosities (i.e., air), tend to be black. Thus, it is possible to observe the relative position of the different components of the facing. Figure 2a shows a front view section of a portion of the clutch lining. The density of the copper is much higher than that of the other components, so it is the material that is most visible in the images and was used to observe the yarn orientation. The network of copper wires can be seen in Figure 2b. The poly-acrylonitrile fibers are also distinguishable as they have a lower density than the matrix. This is because the matrix contains mineral fillers that make its average density close to that of the glass fibers, which are hardly visible.

From the tomography results, it is possible to make cross-sectional views through the thickness at different angles (Figure 2c). Three cuts were made in the tangential direction on the midplane radius (M) and at the level of the inner (I) and the outer (O) radii. The tangential viewing planes show that the copper fibers are elliptical in cross-section, indicating that the fiber axis makes an angle with the direction perpendicular to the cross-sectional surface. This angle corresponds to the angle between the copper fiber and the radial direction (α), which can be obtained through the knowledge of the major and minor axes of the ellipse, *a* and *b* as cos(α) = *b*/*a* [27]. The parameters of ellipses are determined using image analysis (ImageJ software [28]). The average measured angle varies as a function of the radius from 45.7° around the radius *R* = 100 mm (M) to 63° and 68.2° at *R* ≈ 84 mm (I) and *R* = 116 mm (O), respectively. Even if the standard deviation is a little high (about ±10°) due to the fiber waviness (Figure 2b), these values are consistent with the estimate from the theoretical study (Figure 1b). To confirm that the clutch facing can be assumed to be an alternating laminated composite +/−α (angle ply), a cut was made at an angle equal to 50°. Indeed, both angle (α) and angle (−α) provide identical ellipse parameter values. Cutting at this angle shows near-circular sections, meaning near perpendicular cuts, and rectangular sections, associated with longitudinal cuts, which confirms the facing should have an orthotropic behavior close to an alternating layered composite with an average angle near 50°.

### 2.2. Experimental Setup and Method

#### 2.2.1. Digital Image Correlation

The digital image correlation (DIC) technique provides displacements and strains maps on deformed surfaces. The correlation is possible only if the surfaces have a random texture, such as a black and white speckle, which is often obtained with black and white spray paint. A region of interest (ROI) is defined on the specimen surface and is divided into subsets. The algorithm tracks the subsets finding the corresponding location of the reference subsets in the deformed image. To compare the deformed and reference subsets, the DIC algorithm compares the gray level of each pixel in each subset. Optimized criteria for pattern matching have been formulated [29]. Here, the zero normalized sum of square difference correlation criterion is used. Once the matching is performed, the displacement components of the center of the reference and deformed subset can be determined. The strain field is then derived from the filtered displacement field. When using one camera (2D-DIC), the temporal correlation as described above does not take into account out-of-plane displacements. To determine the position of an object in the three dimensions, two simultaneous images of the same object with a different camera angle are needed. Digital image stereo correlation (DISC or 3D-DIC) combines temporal and stereoscopic matching (Figure 3). The DISC method makes the difference between strain and out-of-plane displacements and gives access to 3D profiles. In order to correlate the stereo images, the correlation algorithm needs information on the orientation and position parameters of the cameras as well as the intrinsic parameters of each camera. These parameters are determined by calibrating the stereo-vision setup, which is performed by recording images of a calibration target. The system used in our laboratory is Vic 3D v7, developed by Correlated Solutions (Irmo, SC, USA).

#### 2.2.2. Experimental Setup

Thermal testing was carried out using an XU112 climate chamber developed and specially modified by the manufacturer France Etuves (Chelles, France). To have optical access to the specimen, the climate chamber is equipped with a window on top. The image capturing system consists of two CCD cameras (AVT Pike F-421B, Allied Vision Technologies GmbH, Stadtroda, Germany) with a resolution of 2048 × 2048 pixels providing monochromatic images with 14 bits of dynamic range. The software used in this study (Vic 3D v7) converts the 14-bit images to 8 bits (256 gray levels). The field of view of the cameras is 250 × 250 mm. Therefore, 1 pixel on the CCD sensor corresponds to a 0.12 mm square on the specimen. To illuminate the sample and to limit reflections, the white lights are inside the climate chamber. Two thermocouples are fixed on the rear side of the specimen and in the climate chamber, respectively. The outline of the device is shown in Figure 4. During the calibration procedure, the calibration target is placed inside the chamber so that the cameras view through the window. During the calibration, distortion in the optical path is calculated and taken into account in the algorithm [30]. Thus, considering it is cleaned, the window of the climate chamber has a limited effect on the measures.

#### 2.2.3. Experimental Method

The specimen is painted with aerosol black and white spray paint to create a speckle pattern with adequate contrast. Then, it is placed inside the climate chamber, with no restraints, in a horizontal position. To measure strain due to free thermal expansion, pairs of images are taken at room temperature, defining the reference state of the object. Then, the temperature is increased progressively. Images are taken every 25 °C from 50 to 300 °C, when the specimen temperature is stabilized.

#### 2.2.4. Post-Processing: Resolution and Spatial Resolution

The resolution and spatial resolution of this technique depend on the characteristics of the cameras, the quality of the camera setup, and on the temperature (in chambers). They also depend on the choices made for the post-processing parameters, in particular, the subset size [31]. When processing the data, the subset size was optimized by means of the gray level entropy of the speckle pattern. The spatial resolution of displacement is directly related to subset size. Strain and displacement resolution are determined on non-loaded specimens placed inside the climate chamber at ambient temperature and for each temperature step when the material is stabilized. For a 31 × 31 pixels subset and for this camera setup, the spatial resolution of displacement is 3.7 mm.

In the case of high-temperature measurements, distortion due to the climate chamber window and variation in the refractive index of heated air are sources of measurement errors [32]. To limit the effect of these two sources, the calibration of the stereo system is performed through the window in order to take the additional distortion into account. Furthermore, a fan was placed in front of the climate chamber to homogenize the air between the camera and the window. The strain error was estimated in the optimal conditions by taking images at a steady state for different temperatures from 30 to 300 °C and post-treated with Vic 3D for each temperature independently. In these conditions, the calculated strains are only due to measurement errors. The error is defined as the mean strain of the strain field measured on the surface summed with twice the standard deviation. The strain error versus the temperature is presented in Figure 5a (subset size: 33 × 33 pixels). The strain error increases up to three times its original value. An effective way to reduce the impact of noise when working with digital imaging is to take multiple images in a stabilized state and work on the averaged images (Figure 5a,b).

As shown in Figure 5b, this technique significantly increases the strain resolution over twelve pairs of images. Therefore, in order to reduce the impact of noise, ten pairs of images are averaged for each temperature. These averaged images were then analyzed with the DISC software Vic 3D v7 (Correlated Solutions, Irmo, SC, USA). In the end, the strain resolution varies from 0.01% to 0.05% strain, depending on temperature.

### 2.3. Methods for Determining the Coefficients of Thermal Expansion (CTE)

#### 2.3.1. Coefficient of Thermal Expansion (CTE)

The coefficient of thermal expansion describes material behavior under thermal loading. It is defined, in the simplest form, as the fractional increase in length per unit rise temperature (coefficient of linear thermal expansion). In the case of small deformation, the fractional increase in length is equivalent to the strain (ɛ), and the CTE can be expressed as follows (Equation (2)). The CTE is defined in its linear form over a limited temperature range:(2)$$\mathrm{CTE}=\frac{\varepsilon }{\Delta T}$$
where ɛ is the strain, Δ*T* the temperature variation, and CTE the coefficient of thermal expansion.

For some materials and a large temperature range, it depends on temperature. Especially for polymers, the CTE at the rubbery plateau is much higher than at the glassy plateau, and its measurement versus T is a method to obtain the glass transition T_g_ [3]. Here, the studied temperature range is 30 to 300 °C.

#### 2.3.2. Laminate Theory

As was shown in Section 2.1, the clutch facing may be compared to an angle-ply laminate material with a gradient of change in the angle of the intersection of the fibers (average angle *α* = +/−56°). To validate this approach, the local CTE experimentally obtained was compared to the laminate theory. The intersection angles used are shown in Figure 1b, and the equivalent laminate composite defined by rotational symmetry may be as shown in Table 1 with *L_b_* and *A*_0_ = 40 mm (reference case). The calculations were performed using LAMKIT 1.2 (EADS CCR, Suresnes, France). It is a computation software dedicated to the pre-dimensioning of composite structures, which combines semi-analytical models of analysis of composite laminates and optimization methods [33]. Based on the laminate theory, this software allows the prediction of the behavior of a laminate material subjected to mechanical or thermal loading, depending on the composition of a unidirectional ply and the stacking.

#### 2.3.3. Finite Element Method

A 3D finite element (FE) model (*D_in_* = 160 mm, *D_out_* = 240 mm, depth = 2.55 mm) was performed using ABAQUS/Standard 6.9 software (Dassault Systèmes Simulia Corp., Providence, RI, USA), with default convergence criteria parameters. The shape of the clutch facing is simple, and the hexahedral elements reduce the number of elements, so the clutch facing was meshed using linear hexahedral elements of type C3D8R, and the element size was 2.5 mm. The calculation time was, therefore, short, which made it easy to perform iterative calculations (see below the Levenberg-Marquardt algorithm), even if the model contained about 3600 elements.

The clutch facing is represented as a 3D homogeneous material with elastic properties and a defined thermal expansion coefficient. The objective here was to use a simple model and not to model all components (fiber and matrix) as in the work of [34,35], where different homogenization methods are compared. To estimate the elastic properties, samples were extracted along the tangential and radial directions, and tensile tests were performed. More details are given in the work of [36] and the results in Table 2. An orthotropic linear elastic material model was therefore chosen with homogeneous properties.

Regarding thermal expansion, to differentiate the effect of the fiber organization and the effect of the material shape, three cases were studied: an isotropic material, an orthotropic material, and an orthotropic material with a discrete evolution of the radial coefficient of thermal expansion, the step of property evolution is 5 mm (8 concentric sub-regions) (see Section 3.3.3). In the case of orthotropic behavior only, the thermal expansion was characterized by a diagonal tensor with the coefficients along the three directions (radial, tangential, and thickness). The coefficient of thermal expansion with respect to the thickness was determined by conventional dilatometer tests. The density, thermal conductivity, and specific heat were measured globally by usual techniques (buoyancy method, in-plane conductivity test [37], and DSC, respectively) and chosen constant along the three directions. The values are given in Table 2. Finally, a homogeneous thermal loading was applied in a steady state. There are no mechanical boundary conditions to ensure free thermal expansion.

In order to take into account the effects of the fiber organization and the annular structure, the coefficient of thermal expansion tensor was identified using the finite element model updating (FEMU) method. This method consists of adjusting the thermal expansion properties of the finite element model in order to minimize the difference between the experimental and numerical strain field [10] using a Levenberg-Marquart algorithm. This kind of method was applied for a wide range of material and loading conditions [21,22,23].

The Levenberg-Marquardt method is a standard technique for non-linear least-square problems combining features of the steepest descent method and the Gauss method [38]. The Levenberg-Marquardt algorithm aims to identify the vector $v^{*}$, which minimizes the cost function J(v) (*cf.* Equation (3)) that can be expressed as the sum of squares of non-linear real-valued functions, representing the error between experimental and numerical results.
(3)$$J\left(v\right)=\overset{N}{\underset{n=1}{\sum }}j_{n}^{2}\left(v\right)=\overset{N}{\underset{n=1}{\sum }}{\left(\frac{y^{num}(v)-y^{\exp}}{y^{\exp}}\right)}^{2}$$
where *y^num^* and *y*^exp^ are respectively the numerical and experimental data.
(4)$$J\left(v^{*}\right)=\min\left(J\left(v\right)\right)$$

During the execution of the Levenberg-Marquardt algorithm, the initial parameter guess *v*^0^ is updated such that it minimizes the cost function J(v) according to:(5)$$v^{k+1}=v^{k}+\alpha^{k}d^{k}$$

The search direction *d^k^* can be determined by knowledge of the first and second derivatives of the objective function using the following relations [39]:(6)$$d^{k}=-{\left(\nabla j^{T}\nabla j+\lambda I\right)}^{-1}\cdot \nabla j^{T}j$$
where ∇j is the Jacobian of the vector *j*:(7)$$\nabla j_{\mathrm{mn}}=\frac{\partial j_{m}}{\partial v_{n}}$$

Furthermore, the Jacobian ∇j can be determined by finite differences of that vector:(8)$$\frac{\partial j_{m}(v_{1},\;\ldots ,\;v_{i},\;\ldots ,\;v_{n})}{\partial v_{n}}\approx \frac{j_{1}^{1}\left(v_{1},\;\ldots ,\;v_{i}+\delta v_{i},\;\ldots ,\;v_{n}\right)-j_{1}^{1}\left(v_{1},\;\ldots ,\;v_{n}\right)}{\delta v_{i}}$$
with *δ* = 0.1 in this study.

The damping parameter λ is adjusted in every iteration and initially set to a large value (e.g., 10) corresponding to a variable update according to the gradient descent method [40]. If the iteration results in a worse approximation to the solution, *λ* is increased and otherwise decreased, usually by the factor 10 [41]. The optimization algorithm was developed by coupling the ABAQUS/standard code with a Python script. The Python script () was used to perform all the calculations of the algorithm, modify the parameters to be optimized in the ABAQUS input file, and run the finite element computations. The version of Python installed with Abaqus 6.9 was 2.4.3.

## 3. Experimental Results

This section presents all the experimental results, starting with a validation of the experimental setup on known materials. Then, the analysis of the strain fields of the clutch facing subjected to a homogeneous thermal loading is presented, highlighting the effect of fiber organization. Finally, the three methods for determining the thermal expansion distribution are compared.

### 3.1. Experimental Validation

#### 3.1.1. Validity Tested on Isotropic Materials

In order to verify the validity of the proposed technique, free thermal expansion of pure aluminum (A5) and aluminum oxide (Al_2_O_3_) specimens were measured. The procedure is the same as described in the previous Section 2.2. Mean values of the strains obtained for the two specimens are compared to the results from the corresponding handbooks and presented in Figure 6. The error bars represent the mean strain and standard deviation of the strain field for each temperature. The strain measurements of the Al_2_O_3_ specimen are more affected by errors, as the mean strains are closer to strain resolution. In each case, strain fields were homogeneous and showed suitable agreement with CTE reported in the literature [42,43].

#### 3.1.2. Validity Tested on Transversely Isotropic or Orthotropic Materials

This experimental device was developed to measure the thermal expansion of anisotropic materials such as continuous fiber composites. The software Vic3D gives, for each calculated point, a 2 × 2 matrix with normal and shear strains. The principal strains and principal directions can therefore be deduced. In this case, ε_1_ and ε_2_ are the principal strains, and *U*_1_ and *U*_2_ are the principal strain directions. The convention ε_1_ > ε_2_ was chosen. In the case of orthotropic materials, *U*_1_ and *U*_2_ are the in-plane orthotropic axes. In order to verify the effectiveness of this approach, free thermal expansion of unidirectional (UD) carbon fiber (HR) reinforced bismaleimide (*T_g_* (onset) = 202 °C measured at 10 K/min with DSC STAR 01 module from METTLER TOLEDO) was measured. Classic UD principal directions were found (Figure 7a). The technique successfully determined the orthotropic axes and the transverse thermal expansion behavior (Figure 7b). The non-linearity observed on the CTE_T_ curve as a function of temperature is associated with the glass transition of material from glassy to rubbery state.

### 3.2. Experimental Results on Clutch Facing

The device was validated on known isotropic and anisotropic materials. The same procedure was used to measure the coefficient of thermal expansion of the continuous fiber annular-shaped composite described previously (Section 2.1). A view of the clutch facing with the subset grid numerically superimposed on the disc is shown in Figure 8a. The subset size was 31 × 31 pixels, and the step was 7 pixels. The DISC software VIC 3D was used to compute the Lagrange strain tensor. The strains were derived from the filtered displacement field with a filter box size of 15 calculated points. The principal strains are deduced, as explained in the previous Section 3.1.2. The normed principal strain map (ε_1_) for a thermal loading of Δ*T* = 220 °C is shown in Figure 8b.

#### 3.2.1. Orthotropic Axis

As explained in Section 3.1.2, the principal directions can be determined locally. To determine global orthotropic axes, it is necessary to compare the local principal strain direction with a global coordinate system (Figure 9a). The principal strain directions $\left(U_{1},U_{2}\right)$ are deduced from the strains determined in the Cartesian coordinate system $\left(U_{x},U_{y}\right)$. Due to the annular shape of the composite, the cylindrical coordinate system $\left(U_{r},U_{t}\right)$ seems of interest. The local principal strain directions, determined on the annular disc under the thermal loading, are shown in Figure 8b. To test the coincidence of $\left(U_{r},U_{t}\right)$ and $\left(U_{1},U_{2}\right)$ the angle β between the principal strain direction *U*_1_ and the radial direction of the cylindrical coordinate system, *U_r_* was determined for each calculated point (Figure 9c). Though the angle between the principal and cylindrical directions is not homogeneous on the specimen, it is mainly contained between 0° and 25°, as shown on the histogram (Figure 9b), which confirms the orthotropic behavior of the annular clutch facing. The areas where the angle is much higher (between 90° and 60°) could be related to the modification of the yarn tracing during the preforming and curing operations.

#### 3.2.2. Mean Value of Coefficient of Thermal Expansion and Strain Field

Considering the orthotropic axes of the material, the strains are determined in the cylindrical coordinate system. Figure 10 shows the radial and tangential strain fields of a clutch facing as a function of the temperature.

Five identical clutch facings were tested with the same thermal loading. The mean radial (ɛ_R_) and tangential (ɛ_T_) strain versus temperature variation (Δ*T* = *T* − *T*_0_ with *T*_0_ = 30 °C) are shown in Figure 11. In the temperature range of 30–250 °C, the composite has a linear behavior. Two coefficients of thermal expansion were identified using Equation (2).

Furthermore, Figure 10 and Figure 11 show that the full-field strain data revealed that the radial and tangential strain is not homogeneous throughout the disc and has a specific pattern in the radial direction. The strain appears to be, at first order, radius-dependent (Figure 8b, Figure 9c and Figure 10). The mean strain function of the radius is then determined on concentric rings 5 mm wide (corresponding to the spatial resolution of the experimental device) with a pitch of 2.5 mm, as shown schematically in Figure 12a for a thermal loading Δ*T* = 220 °C. The radial strain is greater on the edges (inner and outer diameter) than in the middle of the annular composite, and the tangential strain increases continuously from the inner radius to the outer radius (Figure 12b). The error bars represent the mean value, maximum and minimum value of a series of five tests. Figure 12c,d shows that the evolutions of the strains along the radius are similar, whatever the temperature. There are two different causes for the evolution of the radial strain and the tangential strain along the radius: (1) the fiber organization and (2) the annular structure. A discussion of these causes follows.

#### 3.2.3. Effect of the Fiber Organization on the Strain Field

Complimentary tests were conducted on clutch facings with different preforming ratios *L_b_* and track width *A*_0_ (Section 2.1.2) to assess that a stacked-layer composite can be used as a first approach to understanding the mechanical behavior of the finished structure. As explained in Section 2.1.2, the preforming ratio and the track width have an effect on intersection angles between layers. When the preforming ratio or the track width is decreased, the mean intersection angle increases, as shown in Figure 13a and Figure 14a. The change in the fiber organization affects the radial and tangential strain fields, as shown in Figure 13b and Figure 14b.

### 3.3. CTE Distribution Determination

In this part, three methods used to determine the coefficient of expansion distribution will be presented. Each has a different objective: the direct method provides a rapid estimation of CTE distribution from experimental data, the laminate theory approach provides a quick prediction, while the finite element method gives *a priori* closest results compared to reality. Finally, the results of the three methods will be compared.

In agreement with the experimental results, it was assumed that the thermal expansion coefficients depend only on the radius and not on the angular position on the clutch facing.

#### 3.3.1. Direct Method

The first approach consists in calculating the distribution of the coefficients of expansion directly with Equation (2), i.e., dividing local strains by the temperature variation. However, this method does not take into account the possible effect of the annular shape of the clutch facing on the strain field. A result of this method directly determined from the strain field distribution of Figure 12b is given in Figure 15, which shows the radial strain (CTE_R_) and the tangential (CTE_T_) coefficient of thermal expansion according to the radius. In the following, this result will be compared to other methods.

#### 3.3.2. Laminate Theory Results

Experimentally, the radial coefficients of thermal expansion are calculated by averaging strain over concentric sections at each temperature. Each section corresponds to a fiber orientation angle +/−*α* (Figure 1b and Table 1), and a CTE is then assigned to this value. To calculate the tangential CTE, the average strain on the entire disk is considered (this point will be discussed in Section 3.3.3). Figure 16 shows experimental results (CTE as a function of orientation angle +/−*α* for the five tested configurations of composites.

From a numerical point of view, the radial and tangential CTE are based on the laminate theory using LAMKIT software (see Section 2.3.2). However, the properties of the fibers and the matrix are not known. The properties of a ply were adjusted in order to estimate exactly the same CTE as in the experimental case for the angle *α* = +/− 56° (Table 1) (red dot in Figure 16). A very suitable correlation between the evolution of the CTE in the function of *α* between the layered model (blue and red lines) and the experimental results (dots) can be seen in Figure 16.

In Figure 17, the radial and tangential strains numerically calculated using the laminate theory are compared to experimental measurements in the case of another preforming ratio (0.8 *L_b_*). Despite assumptions of discrete distribution and a theoretical determination of angles of the intersection, the radial strain field evolution is well described by laminate theory. This approach allows simple and quick prediction of the radial strain field of clutch facings. However, the laminate theory does not explain the evolution of the tangential strain field. Experimentally it increases linearly with the radius, while the laminate theory provides maximum deformation near the median radius. Nevertheless, the magnitude of the estimated strain is suitable. This approach highlights the greater effect of fiber organization in the radial direction.

So far, the modeling was performed without taking into account the annular shape of the clutch facing. To understand the interaction of material properties and structure and its effect on the distribution of the strain field, a finite element model was used. The results are presented in the following paragraph.

#### 3.3.3. Finite Element Method Results

To understand the evolution of the tangential strain with increasing radius, a 3D finite element (FE) model was performed (see Section 2.3.3). As mentioned previously, three cases were studied: an isotropic material, an orthotropic material, and an orthotropic material with a discrete evolution of the radial coefficient of thermal expansion. A homogeneous thermal loading was applied in a steady state (Δ*T* = 220 °C), and the results are shown in Figure 18.

The FE model shows evidence of the effect of the annular structure on the strain field when the material is orthotropic. Despite constant coefficients of thermal expansion, the strain field of an orthotropic material shaped in an annular form presents an evolution along the radius. However, the evolution of the radial strain field, with a maximum strain at the inner and outer radius, is explained by a property gradient.

There are two different causes for the evolution of radial and tangential strains along the radius: the fiber organization and the annular structure. As discussed in Section 3.3.2, with a stacked-layer approach and using the laminate theory, the fiber organization has an important effect on the radial strain. However, this has negligible influence on the evolution of the tangential strain along the radius.

In order to take into account the two effects, the coefficient of thermal expansion tensor was identified using the finite element model updating (FEMU) method (see Section 2.3.3). Abaqus model with an orthotropic material and a discrete evolution of the radial coefficient of thermal expansion was used, with the step of property evolution being 5 mm (8 concentric sub-regions). The goal of this method is to determine the set of coefficients of thermal expansion: ${\left({\mathrm{CTE}}_{R}\left(R_{mean\;i}\right)\right)}_{i=1.8}$ and ${\mathrm{CTE}}_{T}$.

The purpose is to minimize the error between the tangential and radial strains along the radius obtained experimentally ($\varepsilon_{R}^{exp}$ and $\varepsilon_{T}^{exp}$) and numerically ($\varepsilon_{R}^{num}$ and $\varepsilon_{T}^{num}$) for all tested temperatures. The cost function J(v) can be expressed as follows:(9)$$J\left(v\right)=\overset{N}{\underset{n=1}{\sum }}j_{n}^{2}\left(v\right)=\overset{N}{\underset{n=1}{\sum }}{\left(\frac{\varepsilon_{R}^{num}(v)-\varepsilon_{R}^{exp}}{\varepsilon_{R}^{exp}}\right)}^{2}+{\left(\frac{\varepsilon_{T}^{num}\left(v\right)-\varepsilon_{T}^{exp}}{\varepsilon_{T}^{exp}}\right)}^{2}$$

In the numerical implementation of the given problem, the vector *j* is given by:(10)$$j=\left(\begin{array}{c} \frac{\varepsilon_{R}^{num}(v^{k},R_{1},\Delta T_{1})-\varepsilon_{R}^{exp}(R_{1},\Delta T_{1})}{\varepsilon_{R}^{exp}(R_{1},\Delta T_{1})}\\ \ldots \\ \frac{\varepsilon_{R}^{num}(v^{k},R_{12},\Delta T_{1})-\varepsilon_{R}^{exp}(R_{12},\Delta T_{1})}{\varepsilon_{R}^{exp}(R_{16},T_{1})}\\ \frac{\varepsilon_{T}^{num}(v^{k},R_{1},\Delta T_{1})-\varepsilon_{T}^{exp}(R_{1},\Delta T_{1})}{\varepsilon_{T}^{exp}(R_{1},T_{1})}\\ \ldots \\ \frac{\varepsilon_{R}^{num}(v^{k},R_{1},\Delta T_{2})-\varepsilon_{R}^{exp}(R_{1},\Delta T_{2})}{\varepsilon_{R}^{exp}(R_{1},\Delta T_{2})}\\ \ldots \end{array}\right)$$
where $\varepsilon_{R}^{exp}(R_{i},\Delta T_{i})$, $\varepsilon_{R}^{num}(v^{k},R_{i},\Delta T_{i})$, $\varepsilon_{T}^{exp}(R_{i},\Delta T_{i})$, and $\varepsilon_{T}^{num}(v^{k},R_{i},\Delta T_{i})$ are the tangential and radial components of the strain at the radius *R_i_* and for a temperature variation of Δ*T*, as determined from the finite element simulation and the experimental method.

The results are shown in Figure 19 for the reference case (*L_b_* and *A*_0_ = 40 mm), and a very suitable correlation between computed and experimental strains can observe.

## 4. Discussion

Locally, the thermal expansion of the material is not free, as it is bound to the surrounding material. The measured strain fields, therefore, depend on the tensor of the thermal expansion coefficients (i.e., the local orientation of the fibers) but also on the shape of the part.

Figure 20 compares thermal expansion coefficient distributions for the reference case obtained by direct measurement, laminate theory, and finite element method. If the properties of the various constituents of the composite material are known, the laminate theory can provide a quick coefficient of thermal expansion distribution, which can be very important in the case of new product design. For an accurate determination of the deformation fields, the finite element method should be used instead.

Concerning the identification of CTE, the finite element method and direct measurement provide very close results, which means that in our case, the structural effect has a significant influence only on the tangential coefficient.

## 5. Conclusions

The objectives of this work were three-fold: test a new experimental method to measure the thermal expansion of materials using the digital image stereo correlation technique to measure strain fields, apply this method to a continuous fiber-reinforced friction material used in vehicles, and determine the coefficient of thermal expansion tensor.

The experimental setup described in this paper was successfully used to determine the expansion coefficient of known isotropic and anisotropic materials with a strain resolution of 0.01% to 0.05%.

With this new experimental method, the free expansion of a composite clutch lining under homogeneous thermal loading was studied in the 30 and 300 °C range. The strain fields were measured at thermal equilibrium by stereo correlation of images allowing access to the orthotropic axis of the structure. The axes of orthotropy are the axes of the cylindrical reference frame associated with the clutch disc. An inhomogeneity of the deformation according to the radius depending on the axes was shown:Evolution of the radial deformation field with a minimum around the median radius;Linear evolution of the tangential deformation, with a maximum on the external radius.

The extensive information contained in the strain field has also revealed the complex effect of the association of a gradient in the fiber orientation and the annular structure.

The coefficient of thermal expansion tensor was then deduced from the measured strain field with an inverse identification method comparing the experimental strain field and the numerical strain field obtained with a laminate theory approach on the one hand and an FE model on the other.

By inverse identification of the properties of the composite constituents on a reference case, laminate theory allows simple and quick prediction of the coefficient of thermal coefficients and strain fields of clutch facings. This can be a very effective tool for structure design. For accurate identification of the tensor of the coefficients of thermal expansion, the finite element method combined with a Levenberg-Marquardt algorithm allows taking into account structure effect on strain fields and gives better results.

This study was dedicated to the “as-received” material. The behavior of composite materials used in automotive applications can evolve significantly as function thermal cycles, drastic environmental conditions, as shown in the work of [44,45]. Indeed, for the clutch facing, the authors highlighted, using DIC and acoustic emission, the appearance of volume damage in the composite under the effect of thermal fatigue [16]. A paper on thermal aging and its consequences on thermo-mechanical characteristics are currently in progress.

## Figures and Tables

**Figure 1 materials-15-02582-f001:**
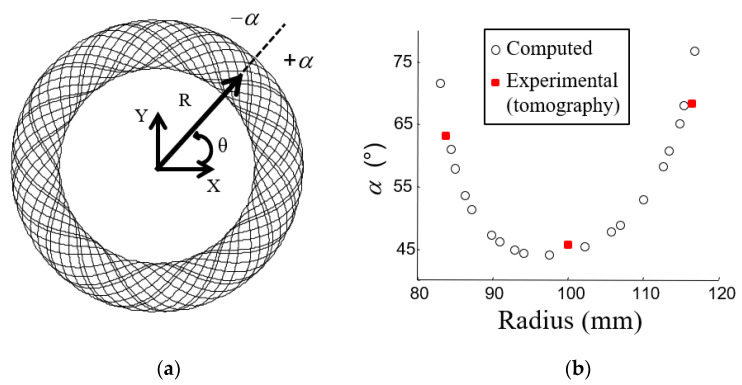
Preform. (**a**) Theoretical fiber organization; (**b**) angle α versus radius (hollow marks: computed values; red dots average values for tomography analysis (Section 2.1.3).

**Figure 2 materials-15-02582-f002:**
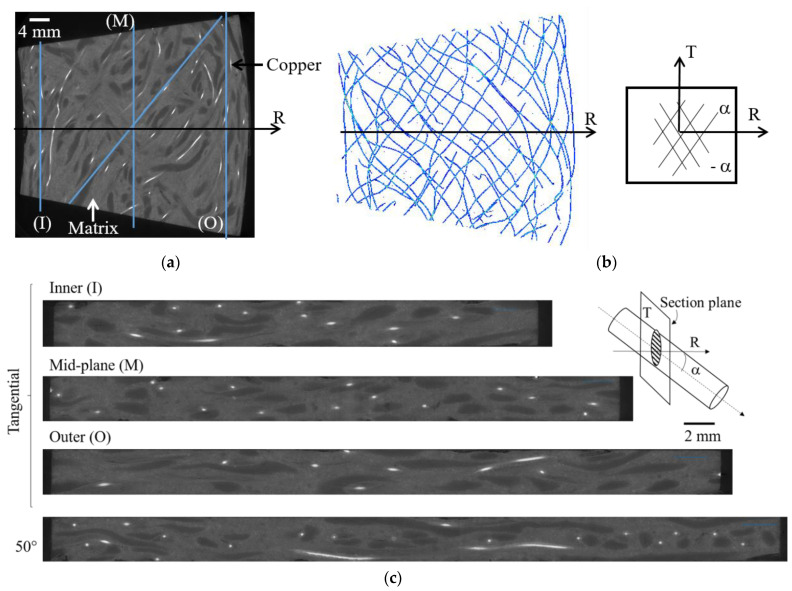
Tomography analysis. (**a**) Front view section of a portion of the clutch lining; (**b**) network of copper wires; (**c**) cross-sectional view in the tangential direction at the inner (I), midplane (M), and outer (O), radii.

**Figure 3 materials-15-02582-f003:**
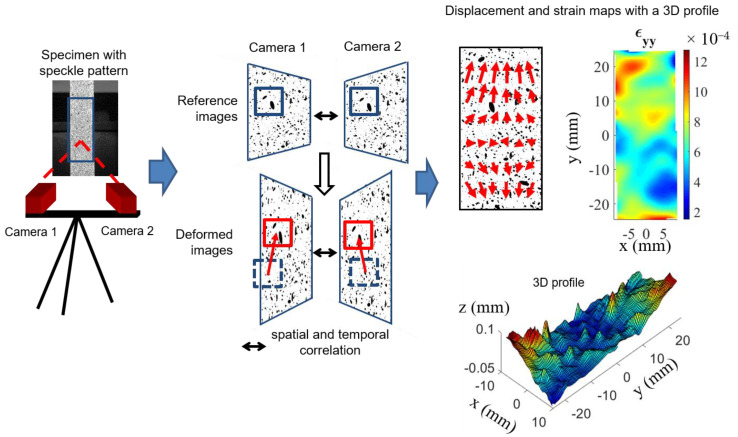
Principle of digital image stereo correlation (DISC).

**Figure 4 materials-15-02582-f004:**
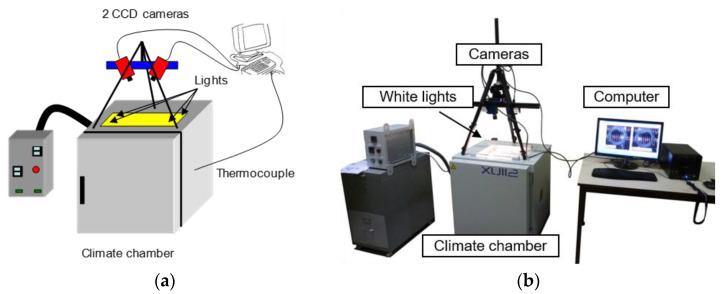
CTE setup. (**a**) Outline; (**b**) photo of the experimental setup.

**Figure 5 materials-15-02582-f005:**
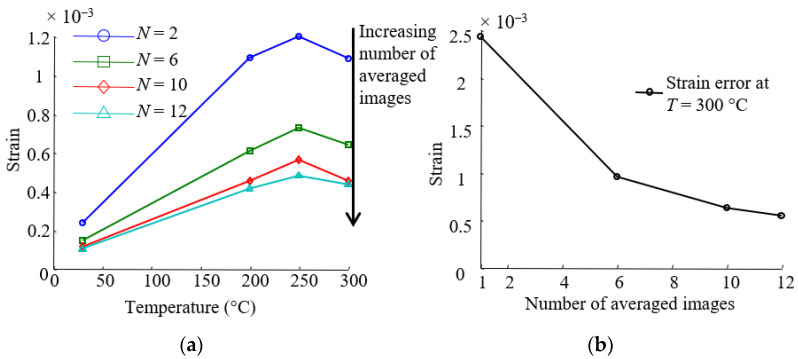
Strain error analysis. (**a**) Strain error versus temperature for different number *N* of averaged images; (**b**) image averaging as a mean to reduce the error: example for strain errors at 300 °C.

**Figure 6 materials-15-02582-f006:**
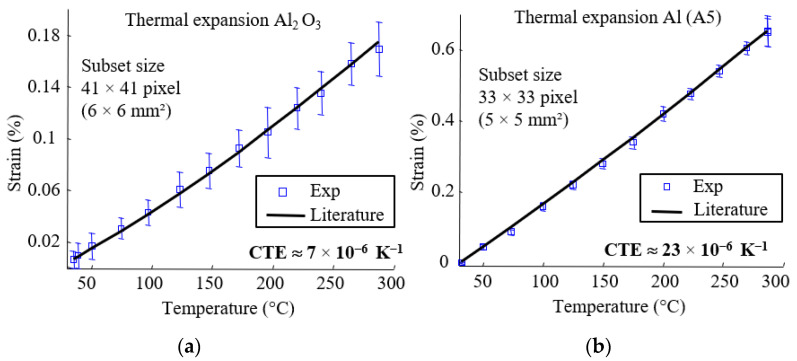
Thermal expansion. (**a**) Aluminum oxide; (**b**) aluminum (A5).

**Figure 7 materials-15-02582-f007:**
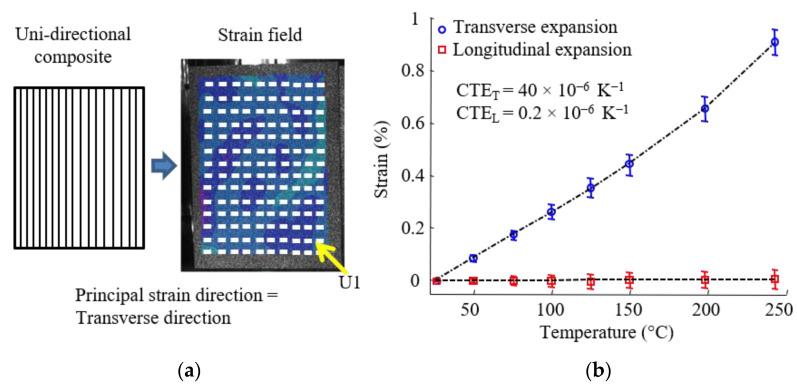
Unidirectional carbon fiber-reinforced bismaleimide. (**a**) Strain field and principal direction; (**b**) free thermal expansion of the composite.

**Figure 8 materials-15-02582-f008:**
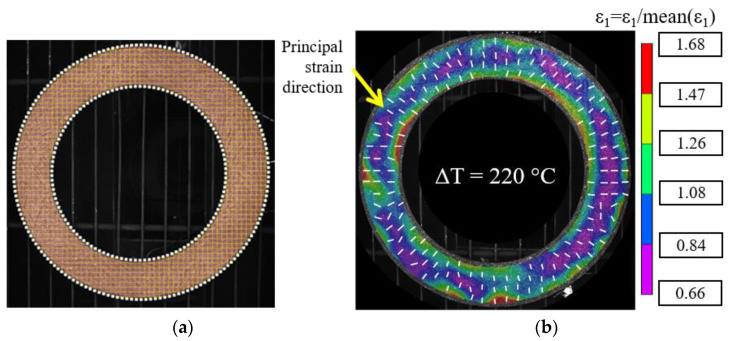
Clutch facing. (**a**) Defined ROI and subset grid; (**b**) measured principal strain field for Δ*T* = 220 °C (*T* = 250 °C).

**Figure 9 materials-15-02582-f009:**
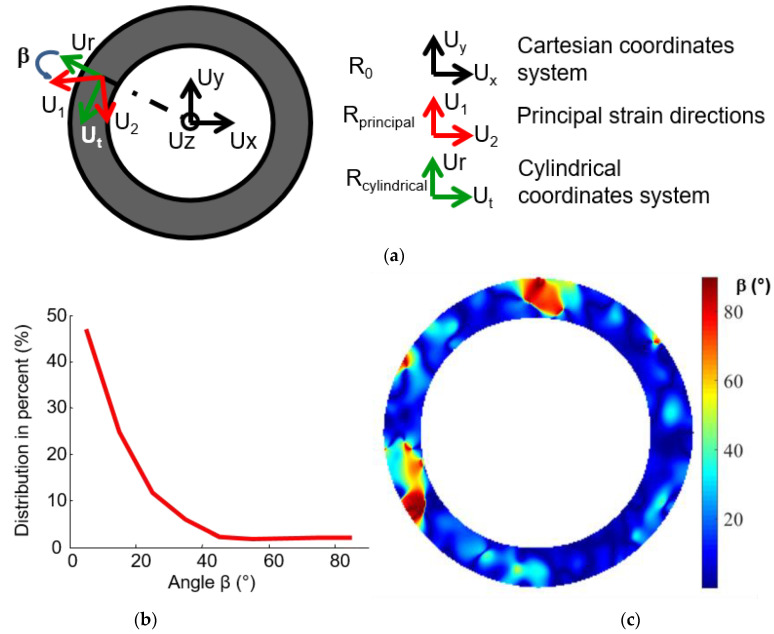
Angle between principal and cylindrical directions. (**a**) Coordinate systems; (**b**) histogram of the angle between the principal and cylindrical directions (β); (**c**) spatial distribution of β.

**Figure 10 materials-15-02582-f010:**
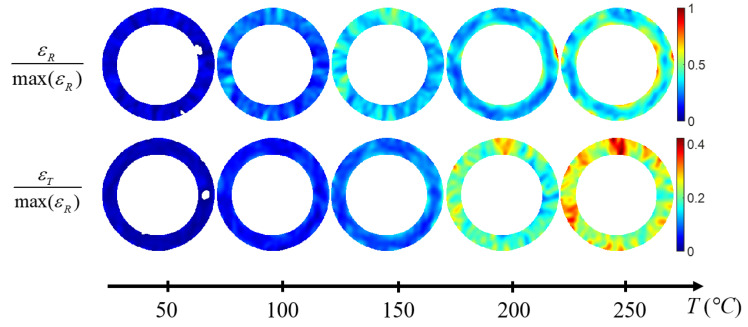
Normed radial and tangential strain fields of a clutch facing as function of the temperature.

**Figure 11 materials-15-02582-f011:**
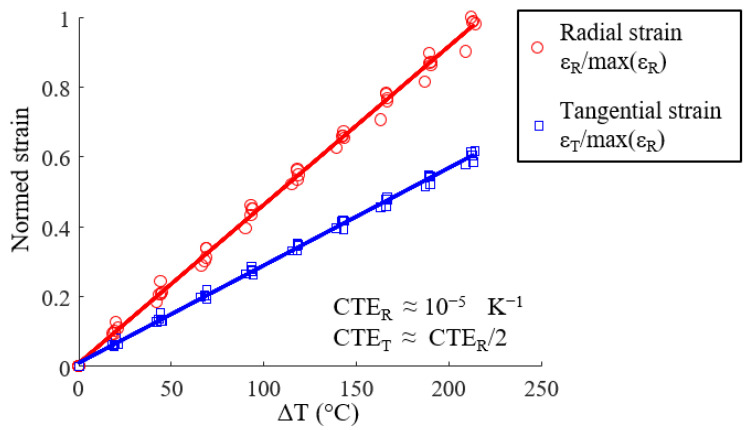
Normed strain versus Δ*T* of five identical clutch facings.

**Figure 12 materials-15-02582-f012:**
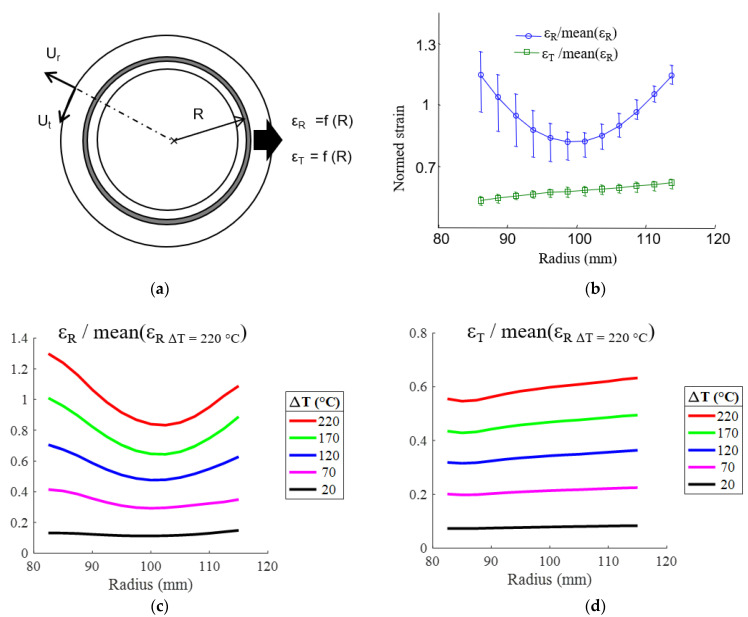
Extraction, from full-field strain, of the radial (ε_R_) and tangential (ε_T_) strains averaged over concentric rings. (**a**) Concentric ring definition; (**b**) strain for a thermal loading of Δ*T* = 220 °C as a function of the radius averaged over five clutch facing; (**c**,**d**) radial and tangential strain as a function of the temperature variation and the radius for a clutch facing.

**Figure 13 materials-15-02582-f013:**
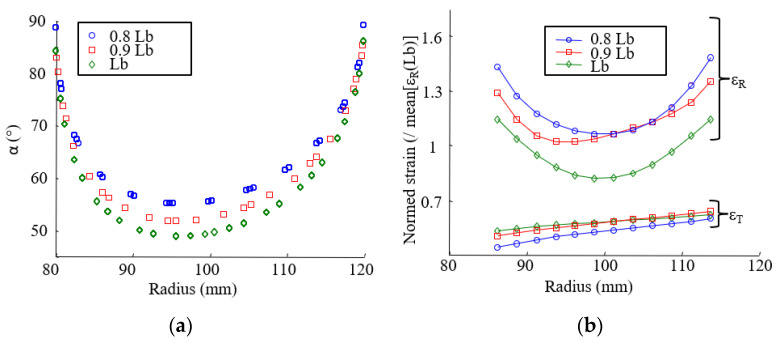
(**a**) Intersection angle versus the radius; (**b**) radial strain (ɛ_R_) and tangential strain (ɛ_T_) versus the radius (Δ*T* = 220 °C) for different preforming ratios (*L_b_*).

**Figure 14 materials-15-02582-f014:**
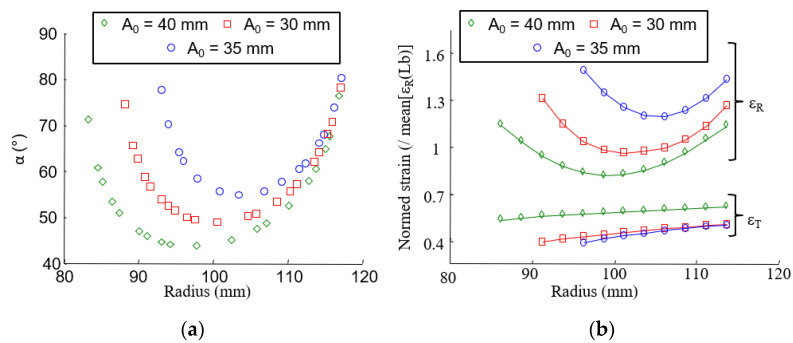
(**a**) Intersection angle versus the radius; (**b**) radial strain (ɛ_R_) and tangential strain (ɛ_T_) versus the radius (Δ*T* = 220 °C) for different track width ($A_{0}$).

**Figure 15 materials-15-02582-f015:**
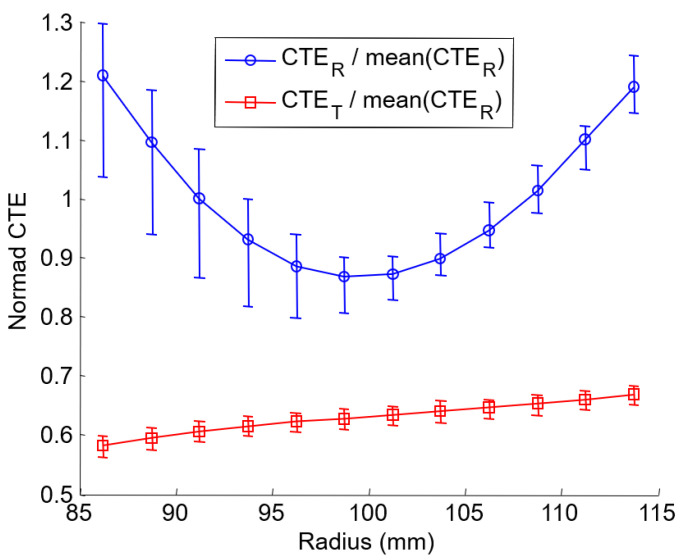
Experimental coefficient of thermal expansion (CTE).

**Figure 16 materials-15-02582-f016:**
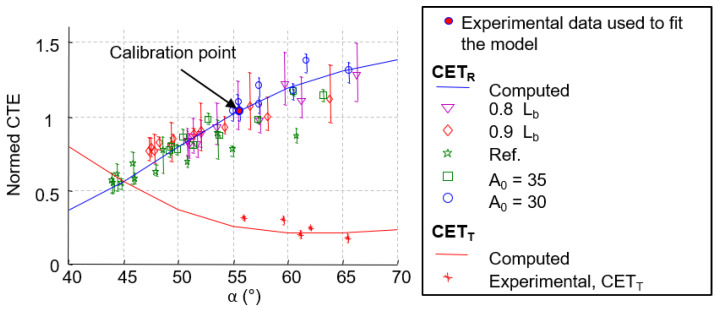
Comparison between local CTE obtained by direct method and the laminate theory.

**Figure 17 materials-15-02582-f017:**
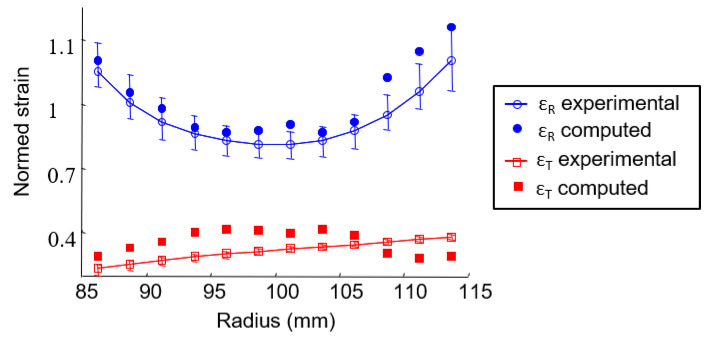
Comparison between experimental and computed strain for preforming ratio 0.8 * *L_b_*.

**Figure 18 materials-15-02582-f018:**
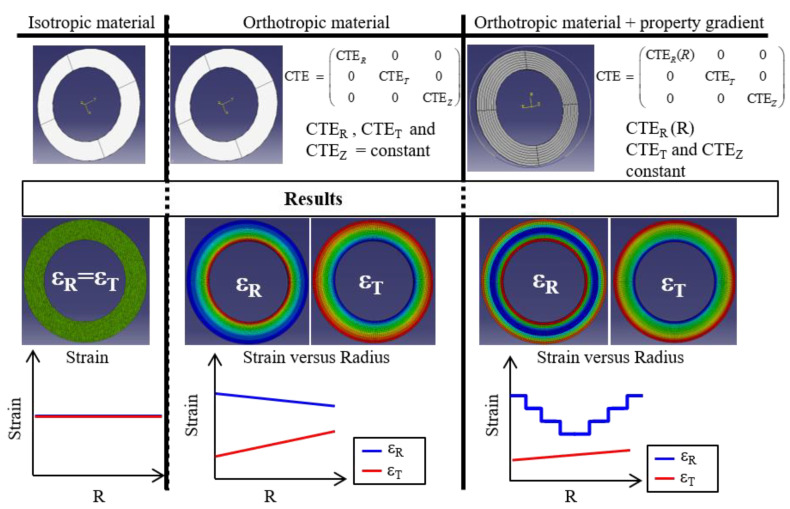
Comparison of the effect of an annular disc and orthotropic behavior. FE models with homogeneous thermal loading (Δ*T* = 220 °C) and free thermal expansion.

**Figure 19 materials-15-02582-f019:**
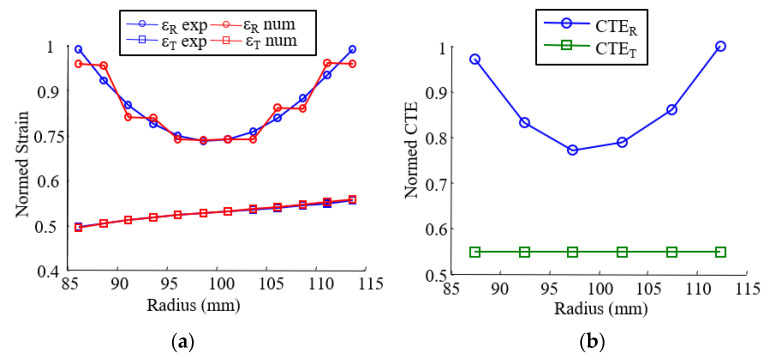
Identification of the material property with an inverse method. (**a**) Experimental and numerical strain after identification; (**b**) normed identified coefficient of thermal expansion.

**Figure 20 materials-15-02582-f020:**
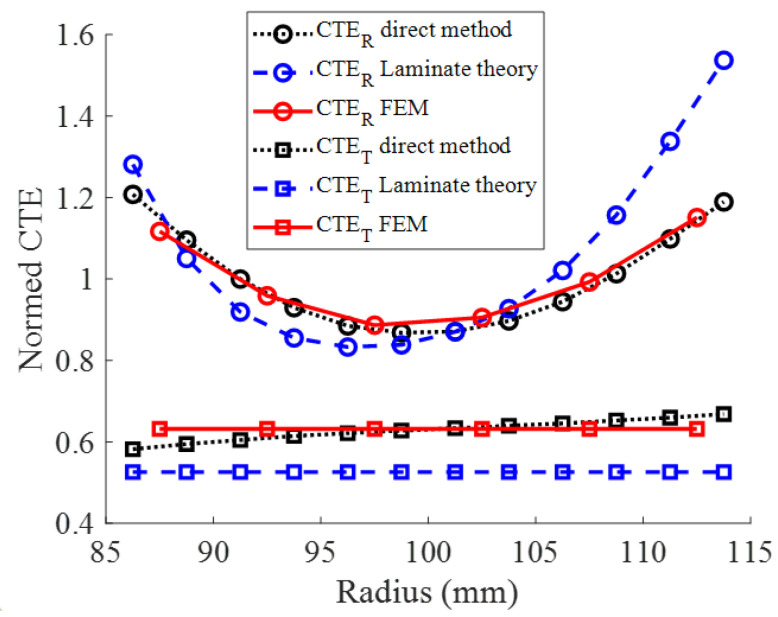
Comparison of CTE estimated by direct measure, laminate theory, and FE method for the reference case (*L_b_* and *A*_0_ = 40 mm).

**Table 1 materials-15-02582-t001:** Equivalent laminate composite for $L_{b}$ and $A_{0}=40\;\mathrm{mm}$ (reference case).

Radius (mm)	α
80–85	+/−67°
85–90	+/−50°
90–95	+/−45°
95–100	+/−44°
100–105	+/−45°
105–110	+/−48°
110–115	+/−56°
115–120	+/−73°

**Table 2 materials-15-02582-t002:** Mechanical and thermal properties of the material.

Tangential Young’s Modulus *E_T_* (GPa)	Radial Young’s Modulus *E_R_* (GPa)	Poisson’s Ratio *ν_RT_*	Shear Modulus *G_RT_* (GPa)	Density (g/cm^3^)	Thermal Conductivity (W/m/K)	Specific Heat (J/g/K)
7.3	4.8	0.45	2.5	1.72	0.3	1.1

## Data Availability

Not applicable.

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
