# Peer review of "Characterization of the Thermal Behavior of a Complex Composite (Clutch Facing) Combining Digital Image Stereo Correlation and Numerical Approach"

_materials, 2022, doi:10.3390/ma15072582_

Round 1

Reviewer 1 Report

This paper deals with the topic on the characterization of the thermal behavior of a complex composite (clutch facing) combining digital image stereo correlation and numerical approach. Experiments, theory and finite element simulations were carried out to obtain the coefficient of thermal expansion and strain field. However, the present writing lacks of the strict logic, which makes it difficult for readers to capture some key results and findings. It is recommended that the authors consider the following comments for the major revision.

  1. In the abstract, it is recommended to simplify the significance of research. At the same time, more qualitative and quantitative research results should be presented, for example the distributions of strain field and thermal expansion coefficient.
  1. The introduction writing lacks of the necessary information. It is suggested that the authors refer to the following specific comments to make relevant supplements.

(1) The car clutch facings are made of continuous glass, polyacrylonitrile (PAN) fibers reinforced polymer composite and copper strips. Therefore, some properties, composition, application and other information about the above composites should be introduced to provide a general understanding for the readers. Furthermore, the performance evolution of the above composites in the high-temperature environment simulating the use process of the car should also be summarized and analyzed. The following related research work can be referred to in terms of composition, application and high-temperature performance of composite. Journal of Materials Research and Technology, 2021, 14:2812-2831. International Journal of Fatigue, 2020, 134: 105480.

(2) Line 49-53, the purpose of the present work should be put after summarizing the research work of others by proposing the unsolved questions.

(3) DIC technology is very effective method to obtain the full field variations of some key mechanical and thermal performances parameters. However, the present summarizing of others’ work is too general, and no specific research work (for example, test accuracy, efficiency and effectiveness etc.) has been reported, which can not provide a very good insight to readers to understand the DIC method. It is suggested that the authors add necessary summary work on DIC method, for example Composite Structures, 2022. 281: 115060. https://doi.org/10.1080/15376494.2021.1974620.

(4) In the present research work, the authors adopt a simple finite element method according to the laminate theory. Therefore, the related research work on using finite element simulation to analyze thermal performance should also be mentioned.

  1. 2.1.1: Please provide the basic parameters related to mechanical and thermal properties for clutch facing material. In addition, it is best to use a flow chart to represent the preparation process.
  1. In the part of materials and methods, the present writing is very chaotic and lacks of the strict logic. It is suggested to adjust the writing according to the materials, the theoretical simulation and experimental testing. The introduction of relevant theories and methods should be placed in the introduction. Part 2.4, Experimental validation should be placed in the results and discussion part. In addition, details of some finite element simulation methods should also be provided. Please rewrite the materials and methods according to the above comments.
  1. Part 3.1 should give specific contents rather than experimental results. In addition, the following secondary headings should be written according to the analyzed content.
  1. Figure 8 should be shown in color to distinguish the changes. In addition, all figures should be presented in high definition.
  1. What is the correlation between the coefficient of thermal expansion and strain field? How to reflect their relationship in the discussion?
  1. Is the maximum temperature of the abscissa in Figure 9 only 220 degrees? Why not analyze the higher temperature? (In the case of repetitive engagement, the temperature rise can be significant: up to 250°C or 300°C depending on clutch technology, which triggers thermal expansion of the material)
  1. Part 3.2.3, the modeling method of finite element should be placed in the materials and methods section. 
  1. The conclusion should be rewritten, including some key information.

Reviewer 2 Report

The work presents methods coupling an experimental device using DIC
and numerical approach to determine coefficients of thermal expansion
(CTE). The authors validated the experimental device with known
materials before its  application to the clutch facing.

The presented results are interesting and well presented. I have some 
minor comments before the final publication of the work.  

1) It will be interesting if  the authors  try to  highlight the limitations of this study.  Suggested improvements of
this work and future directions would be more than welcomed for instance in  the Conclusion section.

2) The authors report that the clutch facing was meshed using linear
hexahedral element of type C3D8R and that the element size was
2.5mm. It would be nice to have more details about the FEM procedure
(number of elements, convergence analysis, possible justification for
the selection of hexahedral elements over tetrahedral elements).

3) As noted by the authors (line 477) the damping parameter λ is adjusted
in every iteration and initially set to a large value. Would it
be possible to report a representative value?

4) In Figures 5 a,b 6b, 11 a,b the * multiplication symbol  should be avoided.

I recommend this paper for publication in Materials.

Round 2

Reviewer 1 Report

The authors have all replied to the comments of the reviewers and recommended to accept the paper.